# Essential Role of COPII Proteins in Maintaining the Contractile Ring Anchoring to the Plasma Membrane during Cytokinesis in *Drosophila* Male Meiosis

**DOI:** 10.3390/ijms25084526

**Published:** 2024-04-20

**Authors:** Yoshiki Matsuura, Kana Kaizuka, Yoshihiro H. Inoue

**Affiliations:** 1Biomedical Research Center, Kyoto Institute of Technology, Mastugasaki, Kyoto 606-0962, Japan; m2641029@edu.kit.ac.jp (Y.M.); cana.j14@gmail.com (K.K.); 2Graduate School of Science and Technology, Kyoto Institute of Technology, Matsugasaki, Sakyo, Kyoto 606-0962, Japan

**Keywords:** COPII, *Drosophila*, male meiosis, cytokinesis, contractile ring

## Abstract

Coatomer Protein Complex-II (COPII) mediates anterograde vesicle transport from the endoplasmic reticulum (ER) to the Golgi apparatus. Here, we report that the COPII coatomer complex is constructed dependent on a small GTPase, Sar1, in spermatocytes before and during *Drosophila* male meiosis. COPII-containing foci co-localized with transitional endoplasmic reticulum (tER)-Golgi units. They showed dynamic distribution along astral microtubules and accumulated around the spindle pole, but they were not localized on the cleavage furrow (CF) sites. The depletion of the four COPII coatomer subunits, Sec16, or Sar1 that regulate COPII assembly resulted in multinucleated cell production after meiosis, suggesting that cytokinesis failed in both or either of the meiotic divisions. Although contractile actomyosin and anilloseptin rings were formed once plasma membrane ingression was initiated, they were frequently removed from the plasma membrane during furrowing. We explored the factors conveyed toward the CF sites in the membrane via COPII-mediated vesicles. DE-cadherin-containing vesicles were formed depending on Sar1 and were accumulated in the cleavage sites. Furthermore, COPII depletion inhibited de novo plasma membrane insertion. These findings suggest that COPII vesicles supply the factors essential for the anchoring and/or constriction of the contractile rings at cleavage sites during male meiosis in *Drosophila*.

## 1. Introduction

Following chromosomal segregation, the dividing animal cells are separated into two daughter cells via plasma membrane ingression, which occurs at the end of cell division. A contractile ring (CR) is first formed at the equatorial cortex of a dividing cell and advances inward to separate into two daughter cells [1]. CR formation is required for the bundled central spindle microtubules at the late meiotic stage [2,3,4]. The Centralspindlin complex is localized in the plus-end region of the bundles of interdigitated microtubules that are found at the cell equator. Centralspindlin localization at the center of the cell induces local RhoA activation. Activated RhoA, in turn, promotes Myosin II activation through the profilin-mediated polymerization of actin and phosphorylation of the myosin regulatory light chain [5,6,7,8,9]. The regulatory factors that determine the cleavage plane are conserved in animal cells, including *Drosophila* and *Caenorhabditis elegans*. The CR components that execute cytokinesis consist of F-actin and Myosin II, and contraction occurs through shrinking of the ring [1]. In addition, the CR contains the cytoskeletal factors septin and Anillin. Both proteins bind to actomyosin rings along the cortex beneath the plasma membrane. Septin polymerizes on its own to form a network structure, and Anillin acts as the master organizer through a subnetwork construct of the CR called anilloseptin, which is RhoA dependent [10]. The CR constructed from these proteins is conserved among all eukaryotes and binds to myosin II and septin [11]. The cleavage furrow (CF) observed during cytokinesis is formed through the ingression of the plasma membrane. In conjunction with CR constriction at CF sites, new membrane components are required. In *Xenopus* eggs, new plasma membranes are inserted into the CF [12], and the membrane components stored in the eggs are reconstituted and transferred to the CF [13]. Additionally, in *Xenopus* eggs, new membrane vesicles are secreted into the base of the advancing furrow via exocytosis [14]. However, the factors required for CR anchoring and new membrane insertion have not been fully elucidated. Intracellular membrane transport is also considered important for plasma membrane ingression in *C. elegans* embryos [15]. Membrane trafficking can be divided into two major pathways—the secretory transport pathway, wherein new membrane vesicles are produced from the endoplasmic reticulum (ER)/Golgi apparatus and transported to the plasma membrane, and the endosomal transport pathway, in which cell surface factors are taken up and returned to the plasma membrane. Both pathways involve several steps, including membrane budding. Budding involves intracellular structures and proteins specific to each step, such as clathrin and the coat protein complex (COP)-coated vesicles, and tethering/fusion-involving receptors (SNAREs) on both the vesicle and target membranes. The depletion of other membrane trafficking factors, such as Arf6, Rab11, and Arf1, also results in the formation of multinucleated mammalian cultured cells [16]. Genetic evidence suggests that these factors are involved in cytokinesis [17,18,19,20,21,22]. COPI-coated vesicles responsible for the retrograde transport from the Golgi apparatus to the endoplasmic reticulum (ER) are required for cytokinesis during *Drosophila* male meiosis [23]. However, there is no clear evidence that anterograde transport of COPII-coated vesicles from the ER to the Golgi apparatus is required for cytokinesis. Even if such evidence existed, it remains to be clarified that COPII-mediated vesicle transport is required at a stage of the cytokinesis process.

*Drosophila* spermatocytes undergoing meiosis I are the largest dividing cells across all developmental stages of *Drosophila*, with a size comparable to that of human Hela cells. This allowed us to perform detailed observations of intracellular structures during cell division. After spermatocytes complete two consecutive meiotic divisions, they form single nuclei and mitochondrial aggregates. This characteristic cell morphology facilitates multinucleated cell detection. Because the spindle checkpoint in mid-meiosis is less strict than that in somatic cells, analysis after late mitosis is possible [24,25]. Owing to these advantages, *Drosophila* spermatocytes are frequently used in studies on cytokinesis. Furthermore, the mutation or silencing of the above-mentioned membrane traffic-related factors in *Drosophila* male meiotic cells results in the same types of cytoplasmic abnormalities as those observed in cultured cells (Arf6 [26], Rab11 [27], Arf1 [23]), suggesting that the role of membrane trafficking factors in cytokinesis is conserved in animal cells [25]. Many membrane traffic-related factors essential for cytoplasmic division have been identified in mutants exhibiting abnormal male meiosis in *Drosophila*. Among them, common phenotypes of an abnormal CR structure and the reversion of plasma membrane ingression have been observed in cells lacking *Rab11* and those with mutant *fws* and *brun*, both of which encode Cog5 orthologs [28,29]. Mutants of the conserved oligomeric Golgi (COG) family, which regulate COPI-mediated transport through the Golgi apparatus, exhibit cytokinesis defects in meiotic cells [28]. COPI is a coatomer complex that coats vesicle and transport proteins from the Golgi apparatus to the endoplasmic reticulum (ER). COPI vesicles, which mediate retrograde vesicle transport, play a critical role in CR formation and supply lipids to the cleavage furrow region during *Drosophila* cytokinesis [23].

In contrast, COPII-coated vesicles encapsulate the proteins synthesized in the ER and transport them to the Golgi apparatus [30,31]. The COPII vesicles are involved in the secretory transport pathway of intracellular membrane trafficking, as described above. COPII is a complex comprising five proteins (Sar1, Sec23, Sec24, Sec13, and Sec31). The small GTPase, Sar1, binds to the ER membrane when converted to active GTP-Sar1. Sec24 binds to the cargo receptor on the ER membrane to incorporate the cargo into COPII vesicles. Another COPII-related factor, Sec16, is present on the ER membrane and regulates COPII formation [32,33,34,35]. Mammalian COPII-coated vesicles also play a role in transporting the bulk of lipids from the ER, which is a significant source of intracellular lipid synthesis [36]. However, this phenomenon has not yet been reported in *Drosophila*. In contrast to the essential role of COPI in cytokinesis, conclusive evidence for the contribution of COPII-coated vesicles to cytokinesis has not yet been reported. Short hairpin RNA (shRNA)-mediated SEC23B suppression in zebrafish erythrocytes has resulted in the production of binucleate cells, indicating a cytokinesis defect [37]. In contrast, because of their large size, detecting defects in cytokinesis in *Drosophila* primary spermatocytes is convenient and sensitive. Although a previous mini-scale RNAi screening to identify essential factors for cytokinesis in *Drosophila* male meiosis among >50 selected genes for membrane trafficking-related factors was performed, genes for COPII-coatomers were not investigated [23]. Clarifying the role of anterograde transport via COPII-coated vesicles in the cytokinesis of *Drosophila* meiotic cells is vital for understanding the final steps of cell division.

Therefore, in this study, we investigated whether COPII-coated vesicles are required for cytokinesis during male meiosis. This is the first evidence of COPII-mediated vesicle transport from the ER to the Golgi apparatus being indispensable for cytokinesis during *Drosophila* male meiosis, which has not been reported or known for any other species. 

## 2. Results

### 2.1. Multi-Nuclear Cells in the Spermatid Cysts Derived from Spermatocytes That Harbored Silencing of mRNAs Encoding COPII Coatomer Proteins

Based on 84 genes, including intracellular vesicle trafficking genes identified from a previous mini-screening, we identified several genes that are essential for cytokinesis during meiosis in male *Drosophila* [23,38]. dsRNA against each target mRNA was specifically induced in spermatocytes according to the GAL4/UAS system using *bam-GAL4*, as well as multinucleated spermatids, which are considered a consequence of cytokinesis failure in meiotic divisions, were examined. We found multinucleate cells in the testes harboring silenced genes (*EXO84*, *Syntaxin5*, *Rab5*, *Rab11*, *Rab19*, *efTuM*, *Sar1*, *Sec13*, *Sec23*, *Sec31*, and *Sec16*) among those involved in intracellular vesicle transport. Of these, the COPII-related genes *Sar1*, *Sec13*, *Sec23*, *Sec31*, and *Sec16* have not yet been investigated to determine whether they are required for cytokinesis during meiosis. Initially, we induced dsRNAs against mRNAs of the above-mentioned five genes and two other genes for a COPII component, Sec24AB and Sec24CD, in primary spermatocytes using *bam-Gal4* and the relevant *UAS-RNAi* stocks that efficiently silenced the relevant mRNAs in male germ cells. We then examined whether multinucleated spermatids in the testes expressed each dsRNA in premeiotic spermatocytes. In total, 16 normal spermatocytes underwent two meiotic divisions to form 64 spermatids. The resulting spermatids possessed a one-to-one ratio of nucleus-to-mitochondrial aggregates after the completion of meiosis (Figure 1A). In contrast, when dsRNA against the mRNA for every five COPII components was induced in spermatocytes before meiosis, multinucleated spermatids were observed immediately after the completion (Figure 1B–G). COPII assembled into two parts—an inner layer, consisting of Sec23, Sar1, and Sec24AB or Sec24CD, and an outer layer, comprising Sec13 and Sec31, surrounding the inner part. Multinucleate spermatids containing two or more nuclei derived from spermatocytes harboring dsRNAs against each mRNA of the six COPII genes were observed at considerable frequencies (Figure 1H). The cytokinesis phenotype was observed in the spermatid cysts of *Sec24CD*-silenced spermatocytes at similar frequencies. In contrast, the multinucleate spermatids were not observed among the *Sec24AB*-silenced spermatocytes (*bam>Sec24ABRNAi*) (n > 190), although the mRNA levels of the gene were reduced to 11% of those of the control. To confirm the results of the silencing experiments, we generated germline cell clones homozygous for a lethal allele (*Sec24AB^k06204^*) of the gene in the testes heterozygous for the chromosome carrying the mutation and the wild-type allele linked to the RFP marker by inducing somatic recombinations. All spermatids homozygous for the mutation exhibited normal nuclear and Nebenkern organization (34 of 34 cells examined). These data suggest that *Sec24AB* is dispensable for cytokinesis during meiotic division.

### 2.2. Sar1-Dependent COPII Foci Containing Sec13 and Sec23 or Sec31 in Spermatocytes before and during Male Meiosis 

Next, we confirmed whether vesicles containing COPII components were formed in spermatocytes and investigated their distribution in the cytoplasm before and during meiosis. First, to visualize the vesicles containing Sec13 and Sec31, we performed the immunostaining of spermatocytes expressing red fluorescent protein (RFP)-tagged Sec13 and green fluorescent protein (GFP)-tagged Sec31. In the interphase cells before meiosis I, an average of 20 foci were evident, which were 0.36 μm^2^ in size per spermatocyte at later growth stages before meiosis I (Figure 2A,I,J). Most Sec31-positive foci overlapped. 

After metaphase I, both foci became smaller and accumulated around sister nuclei (Figure 2B–D). Consistently, most Sec31 foci overlapped with Sec13 foci, indicating that these two proteins co-localized in every focus. To confirm whether these two proteins were closely associated with each other to form a complex in the cytoplasm, we performed PLA in situ on premeiotic spermatocytes expressing Sec13-RFP using anti-RFP and anti-Sec23 antibodies (Figure 2E,F). The PLA signal appeared on all of the Sec13-positive foci in the spermatocytes at the S5 stage before meiosis (Figure 2E’) (average: 60.1 foci, n = 44 cells), whereas few PLA in situ signals appeared in the cells when only an anti-RFP antibody was used (Figure 2F’) (average: 2.9 foci, n = 51 cells). These results suggest that these two COPII coatomer proteins are localized close enough to form COPII-coated vesicles in the cytoplasm of spermatocytes.

Next, we confirmed that COPII-containing foci were constructed based on the assembly factor Sar1. We observed the fluorescence of RFP-Sec13 and GFP-Sec16 in the control and *Sar1*-scienced spermatocytes expressing these fluorescent proteins (Figure 2G,H). The number of Sec13-positive foci decreased to one-third of that in the control cells in *Sar1*-silenced cells at late anaphase I (Figure 2K), and the fluorescence intensity was reduced (Figure 2G,H). These observations strongly suggested that COPII vesicles containing Sec13, Sec16, or Sec23 are formed in a Sar1-dependent manner in spermatocytes before and during meiosis.

### 2.3. Co-Localization of COPII-Containing Foci with cis-Golgi Marker and Their Dynamic Astral Microtubule-Dependent Distribution in Meiotic Cells

Next, to understand the role of COPII-coated vesicles in cytokinesis, we investigated the intracellular localization of COPII foci during meiosis I progression. RFP fluorescence of living and fixed spermatocytes expressing RFP-Sec13 was observed (Figure 2A’ and Appendix A). The number of COPII-containing foci increased ten-fold until metaphase I. Accordingly, the foci halved in size at anaphase I and thereafter (Figure 2J). Subsequently, COPII-containing foci accumulated in the two cytoplasmic bands running from the top to the bottom of the cell, which were more centered in each sister nucleus than that at the late anaphase (arrowheads in Figure 2G,G’). The foci concentrated around both sister nuclei during the telophase (Figure 2D). However, COPII-containing foci did not accumulate at the CF sites on the plasma membrane (Figure 2D,D’’). Until prometaphase I, all COPII foci overlapped with GM130-positive vesicles in spermatocytes immediately before meiosis (Appendix A). These vesicles corresponded to tER-Golgi units. Most Sec13-positive vesicles overlapped with GM130-positive vesicles. These observations suggest that the foci containing RFP-Sec13 and anti-GM130 signals correspond to the tER-Golgi units and tER sites, where the COPII vesicles budded. During male meiosis, the number of cis-Golgi marker foci decreased in *Sar1*-silenced spermatocytes (Appendix A). COPII and cis-Golgi reduced in size and fluorescence intensity (Appendix A) at and after anaphase I. 

To elucidate the mechanism underlying the dynamics of COPII-containing tER-Golgi units, we investigated whether microtubules emanating from the spindle poles were involved in their distribution. A time-lapse observation of microtubules and RFP-Sec13 vesicles simultaneously showed that the vesicles appeared to cluster on the plus-end regions of the astral microtubules to avoid the pole regions (Appendix A). To confirm their microtubule association, we investigated whether testis-specific *β2tubulin* and hypomorphic *asterless* mutations influenced foci distribution. In the spermatocytes heterozygous for a dominant mutation of testis-specific tubulin gene (*β2Tub85D^D^/+*), 77.3% (n = 66 cells) of the mutant cells had an abnormal COPII vesicle distribution. In comparison, only 4.5% of the control cells (*+/+*) had an abnormal COPII vesicle distribution (Appendix A). Consistently, all the spermatocytes examined (n = 84 cells) in hypomorphic *asl* mutant (*asl/asl^2^*) males exhibited an abnormal COPII distribution, whereas only 6.5% of the control cells (*asl^2^/+*) showed this phenotype (Appendix A). These results are consistent with the interpretation that astral microtubules emanating from centrosomes are involved in the dynamics of COPII-containing tER-Golgi units during male meiosis. 

### 2.4. Sar1 Silencing Did Not Affect Furrowing Initiation

COPII-containing vesicles are formed in spermatocytes and are required for cytokinesis during meiosis. This led us to investigate the role of COPII-mediated intracellular transport in cytokinesis. To understand which process(es) of cytokinesis were affected in the *Sar1*-silenced spermatocytes and the mechanism whereby COPII-mediated transport is required for cytokinesis in male meiosis, we performed a live analysis of the cells undergoing meiosis I to observe plasma membrane ingression. First, we labeled the plasma membrane with GFP by expressing the GFP-tagged protein localized on the plasma membrane (PLCγ-PH-GFP) and observed the membrane dynamics in a time lapse. Therefore, we examined whether COPII silencing resulted in a similar defect in cytokinesis. In control meiotic cells, plasma membrane ingression began in the late stage of meiosis I (t = 0′) (Figure 3A). Twenty-five minutes after the start of furrowing, plasma membrane ingression no longer progressed. The midbody structure connecting the daughter cells remained and stabilized as a ring canal (t = 29′) (Figure 3A). The distance between the furrowing CF edges was gradually narrowing after the furrow initiation (Figure 3C). In *Sar1*-silenced cells, plasma membrane ingression was initiated, as observed in the control cells. Furrowing appeared to progress properly until 15 min after initiation (Figure 3B). However, the ingression was arrested in the middle of cytokinesis (t = 31′) and returned to the original state before the initiation of cytokinesis at the end (Figure 3B) (t = 39, 47, 56′). In other *Sar1*-silenced cells, the position of the CF shifted from the center to the left-hand end of the cell (two out of six cells showed abnormal cytokinesis). Afterward, the plasma membrane ingression was arrested and eventually returned to the original state, as described above (t = 56′) (Figure 3D). All the control cells (15 of 15 cells examined) completed cytokinesis properly, whereas >40% (6/13) of the *Sar1RNAi* cells showed abnormal plasma membrane ingression, and eventually, meiotic divisions were perturbed. These observations indicate that COPII silencing results in the abnormal ingression of the plasma membrane rather than the initiation of cytokinesis, unlike the phenotype observed in COPI-silenced spermatocytes.

### 2.5. Sar1 Silencing Led to Abnormal Contractile-Actomyosin and Anilloseptin Ring Localization at the Cell Equator in Telophase I

To understand the mechanism of cytokinesis failure in *Sar1*-silenced meiotic cells, we investigated whether CRs executing plasma membrane ingression were localized at CF sites. In control cells (*bam>+*), contractile actomyosin rings were formed of myosin light and heavy chains, and F-actin rings were formed in control telophase I cells (Figure 4A,B). In contrast, neither Myosin II nor F-Actin rings were properly localized at telophase I in the *Sar1RNAi* cells, unlike in the controls, although a part of each ring remained localized at only one side of the CF regions (Figure 4E–E’’,F,F’). Consistently, the contractile rings consisting of Anillin and Septin 1 were also mislocalized in the RNAi cells at telophase I (Figure 4G,G’,H,H’).

Next, we investigated whether these rings were once formed and removed from one side of the plasma membrane or whether the ring components abnormally accumulated only on one side of the CF in COPII-depleted telophase I cells. Thus, we performed time-lapse observations of Myosin Light Chain (MLC)-GFP fluorescence during the later stages of meiosis I in the control (*bam>+*, Figure 4I) and *Sar1*-silenced cells (*bam>Sar1RNAi*, Figure 4J). Once the MLC ring was correctly formed, the furrowing initiated (t = 28′) in the silenced cell. However, when GFP fluorescence disappeared at the lower CF site, furrow ingression returned to the initial state of the plasma membrane (t = 48′). Consistently, the anilloseptin rings consisted of Septin1 (Figure 4K,L,O), and Anillin (Figure 4M,N,P) was formed at the CF sites and removed from one side of the CF site as the rings were constricted (Figure 4K,M for the control cells and Figure 4L,N for the silenced cells). In summary, both actomyosin and anilloseptin subrings were once formed and removed from the plasma membrane upon contraction.

### 2.6. Sar1-Dependent Formation of Vesicles Containing DE-Cadherin in Spermatocytes and Their Accumulation in the Cleavage Furrow Sites during Male Meiosis

Because CRs were initially formed even in *Sar1*-silenced cells, we speculated that the rings became unstable in COPII-depleted cells after their temporal construction. We suspected that the COPII-mediated anterograde transport of factor(s) that anchor the CR to the plasma membrane would be disrupted in COPII-silenced cells. Here, we focused on the cell adhesion factor E-cadherin (DE-Cad) as a candidate for stabilizing the CR because it penetrates the plasma membrane and binds to F-actin through the β-catenin encoded by *arm*. Therefore, we considered the possibility that DE-Cad may play a role in stabilizing the CR at the cleavage site. Accordingly, we examined whether DE-Cad vesicle formation and their localization at the cell equator depended on the COPII-mediated transport in meiotic cells. In normal control cells, DE-Cad accumulated at telophase I (Figure 5A,B), whereas in *Sar1*-silenced cells, fewer vesicles contained the protein in interphase (Figure 5C) and meiotic cells (Figure 5D). An average of seven vesicles per cell was observed in the controls (n = 58). In contrast, an average of 1.2 vesicles were found in *Sar1*-silenced cells (n = 62); this difference is statistically significant (Figure 5G). 

Next, we performed a time-lapse observation of the GFP-DE-Cad fluorescence during meiosis and recorded the images from early anaphase I (t = 0′) to the end of cytokinesis (t = 27′) (Figure 5E). DE-Cad-containing vesicles migrated from the periphery of the spindle poles toward the center of the cell at metaphase I. Subsequently, DE-Cad-containing vesicles accumulated at the CF sites so that the vesicles fused with the existing larger ones (Figure 5E,E’) (t = 27′). In contrast, few DE-Cad-containing vesicles, except for those uniform and with less intense GFP fluorescence, were observed in meiosis I cells (Figure 5F). These observations suggest that these DE-Cad vesicles are intracellular transport system dependent. 

### 2.7. COPII Depletion Disrupted Novel Membrane Insertion into CF Sites during Cytokinesis

We further investigated whether COPII contributes to new plasma membrane insertions at the cleavage furrow sites. To this end, wheat germ agglutinin (WGA) can be used to label newly synthesized plasma membranes, because it binds to sialic acid-containing glycoproteins. Using this marker, we investigated whether newly added plasma membranes at the CF sites were affected in *Sar1*-silenced cells. In control cells, the intense WGA signal accumulated around the CR was visualized through the Anillin in the furrowing plasma membrane at the midzone of telophase I cells (Figure 6A). In contrast, in *Sar1*-silenced cells, weak WGA signals were spread over the entire plasma membrane and were not observed in the newly added furrowed plasma membrane (Figure 6B). Abnormal cells exhibited no or reduced WGA signals in the midzone among telophase I cells without furrow ingression (n = 36 control and silenced cells). WGA signals were not detected on the new membrane around the CR in 88.9% of the cells exhibiting regression at telophase I (Figure 6C), whereas only 20% of control cells had no or reduced WGA signals. This suggests that COPII silencing perturbs the new plasma membrane in addition to the CF sites.

## 3. Discussion

### 3.1. COPII-Coated Vesicles Are Indispensable for Cytokinesis in Drosophila Male Meiotic Division 

The co-localization of Sec13 and Sec31, which constitute the inner layer of COPII, and a close association of Sec13 with Sec23, a component of the outer layer of COPII, are evident from our findings. These data strongly suggest that COPII vesicles were constructed in the cytoplasm of spermatocytes. In addition, Sec13 co-localizes with Sec16, which is localized on the ER membrane and with cis- and mid-Golgi markers. Sec13-positive foci did not perfectly overlap with the trans-Golgi marker, which suggests that COPII components co-localized with the tER-Golgi units but were slightly apart from the trans-Golgi network. These observations are consistent with previous findings in that COPII vesicles exist between the tER and Golgi cisternae or are closely linked to ER-Golgi units [39]. Thus, the COPII foci visualized using RFP-Sec13 may correspond to the tER-Golgi units from which COPII budded. 

The genetic data indicate that the silencing of six COPII components, including Sar1 and Sec16, in spermatocytes results in the production of multinucleated post-meiotic cells, indicating that COPII-mediated intracellular transport is required for cytokinesis in male meiosis I. In contrast, silencing *Sec24AB*, an outer layer component, did not produce multinucleated cells in either or both meiotic divisions, while that of its alternative gene, *Sec24CD*, resulted in multinucleated spermatid production [40]. None of the spermatocytes homozygous for the lethal mutation of *Sec24AB* produced multinucleate cells. The *Sec24AB* gene is dispensable for cytokinesis in male meiosis, although it is expressed at high levels in the testes (https://flybase.org/reports/FBgn0033460: accessed on 1 April 2024). In human cells, proteins synthesized in the ER comprise amino acid sequences called export signals, which Sec24 recognizes. COPII forms capsular vesicles that internalize and export cargo proteins from the ER [41]. Sec24CD, but not Sec24AB, may be used to select cargo proteins (s) essential for cytokinesis in *Drosophila*. The pre-cis-Golgi region, which corresponds to the ER-Golgi intermediate region (ERGIC) in vertebrates, is involved in protein exchange between the ER and Golgi apparatus in *Drosophila* as well as in anterograde and retrograde trafficking. Furthermore, a tube-like structure, directly connecting the ER exit sites (ERES) and the Golgi apparatus, was recently observed in this region. This suggests that proteins synthesized in the ER are directly transported through this tube to the Golgi apparatus [39]. We showed that the COPII components co-localized closely with each other to form the same complex. Our data suggest that DE-Cad and Septin1 are candidate cargo proteins involved in cytoplasmic transport. 

### 3.2. COPII-Coated Vesicles Are Essential for the Continuous Ingression of the Cleavage Furrow While Coordinating with CR Constriction in Cytokinesis

Myosin II and F-actin, consisting of the CR, as well as Anillin and Septin1, components of the anilloseptin ring, were dissociated from the plasma membrane in CopII-depleted cells after the normal formation of the rings and initial ingression of the CF. Anillin is required for the recruitment of septin to the CF and maintenance of actomyosin at the equator during cytokinesis [42,43,44]. Thus, the regression of the furrowing plasma membrane in depleted cells may result from the supply of protein(s) connecting the CR to the membrane, and the formation of new membrane components in the cleavage sites may be inhibited. F-BAR domain proteins that change the curvature of the plasma membrane and anchor the CR to the plasma membrane, such as Syndapin, are candidates for proteins that are conveyed to the CF sites via COPII-mediated transport [45]. Unfortunately, we did not succeed in visualizing Syndapin in the plasma membrane of meiotic cells. It remains unclear which cargo proteins of COPII are directly involved in the abnormal dynamics of the CR. 

The cleavage plane was determined using Rho prior to cytokinesis, which triggers CR formation in HeLa cells [46,47]. Since COPII silencing did not affect the accumulation of CR components, COPII may not affect the localization of ring components. However, we observed that the cleavage furrow was off the center of the COPII-depleted cells. Similar abnormalities have been observed when Pavarotti kinesin-like protein (PAV), a component of the chromosomal passenger complex (CPC) and the Centralspindlin, is silenced [25]. Microtubule misalignment was observed in the silenced cells, leading to CR formation in some cells that were displaced from the center. This raises the possibility that COPII is also involved in the cellular localization of CPCs and PAVs. 

### 3.3. Sar1-Dependent DE-Cadherin-Containing Vesicles Accumulate around the CR 

As DE-Cad-containing vesicles are formed in a *Sar1*-dependent manner in spermatocytes, this protein is a candidate cargo protein of COPII. DE-Cad binds to neighboring cells to form adherens junctions through its extracellular domain. The intracellular domain binds F-actin through the backbone protein catenin [48,49]. We observed the co-localization of DE-Cad and β-catenin at late stages of meiosis I. This is consistent with a previous finding in that DE-cad recruits α- and β-catenins to the CF and stabilizes the actomyosin CR [42]. Our time-lapse observations of DE-Cad demonstrated that the vesicles migrated from the periphery of the spindle poles toward the CR and ring canals. Vesicle formation depends on Sar1 in meiotic cells. These results suggest that DE-Cad is transported via COPII vesicles. In human epithelial cells, extracellular E-cadherin is internalized into vesicles via endocytosis together with p120 catenin [50,51]. In contrast, DE-Cad transport toward the CF is mediated by vesicles other than COPII because COPII is not localized to the CF. COPI, which is responsible for retrograde transport, did not accumulate in the CF [23]. In contrast, Rab11 in *Drosophila* meiotic cells and Rab35 in *Drosophila* S2 cells accumulate in the cleavage furrows (CF)s [27,52]. These Rab-mediated vesicles, which are involved in recycling endosomes, are candidates that directly transport DE-Cad to the CF. In mammals, N-cadherin is transported to the murine cerebral cortex via Rab11 or Rab5 [53]. Therefore, it is necessary to examine the relationship between DE-Cad and recycling endosomes in future studies.

Cadherin is a transmembrane protein that forms adhesion junctions with neighboring cells. DE-cad organizes an abundant adherence junction between germline stem and hub cells in the *Drosophila* testis apical niche. Ectopic DE-Cad expression lacking the extracellular domain prevents centrosome and spindle orientation of germline stem cells, indicating that DE-Cad is essential in the niche [54]. However, whether DE-Cad is required in *Drosophila* meiotic cells needs to be further investigated.

### 3.4. Involvement of COPII Vesicles in the Supply of Membrane Components to Newly Added Plasma Membrane Connected to the Constricting CR

A notable aspect of cytokinesis failure due to COPII depletion is the observation that the plasma membrane reverts to its original state once cytokinesis is initiated. However, this phenotype is not observed in COPI-deficient cells [23]. This regression was observed at approximately the same time as was the dissociation of the CR from the plasma membrane. This indicates that regression occurs because of the loss of connection between the ring and the membrane. Anillin, a component of CR, binds to the plasma membrane component PI(4,5)P2 via F-BAR domain proteins such as Syndapin [45]. These previous findings suggest that the supply of specific lipid components to the CF site is essential for the interaction between the CR and the membrane. The depletion of COPI results in a loss of lipid droplet storage, suggesting that COPI vesicles transport lipids to storage sites [23]. In mammalian cells, membrane lipids are synthesized in the ER and Golgi apparatus, and COPII directly mediates the intracellular transport of these lipid components [55]. An insufficient supply of lipids to the CF results in the failure of cytokinesis during animal mitosis. Once invaginated, the cell membrane reverts to its original state midway through cytokinesis [56]. We showed that newly added plasma membranes at the cleavage plane during cytokinesis remarkably reduced the frequency of telophase cells with new membranes in COPII-depleted cells. Therefore, it is important to investigate whether the COPII vesicles are also involved in transporting membrane lipid components in male meiosis. Since COPII was not localized to the CF in *Drosophila* meiotic cells, the vesicles may be indirectly involved in lipid transport toward the CF, even if the lipid accumulation would be inhibited in COPII-depleted cells. 

### 3.5. Comparing the Role of Coatomer-Mediated Transport in Mitosis and Drosophila Male Meiosis and between COPI- and COPII-Mediated Transport in Male Meiosis

Protein transport between ER and Golgi by COPI and COPII is suppressed during the M phase of the cell cycle in mammalian cells. The small GTPases, Arf1 and Sar1, that regulate vesicle assembly are switched off at the onset of mitosis and reactivated as the cells exit mitosis [17]. Each GTPase regulates the conformational changes in the ER and Golgi as the cell cycle progresses. When Sar1 is downregulated, COPII components can no longer accumulate on the ER membrane. The ectopic expression of the constitutively active Arf1 inhibits not only the Golgi transformation, but also chromosome segregation and cleavage furrow formation during cytokinesis in mammalian cells [17]. By contrast, a previous study demonstrated that COPI components and Arf1 were required for *Drosophila* male meiosis [23]. Our current study also showed that COPII-mediated anterograde protein transport between ER and Golgi is required for male meiosis. These findings suggest that the requirements for the COPI and COPII-mediated vesicle transport requirements may differ between cell types. The *Drosophila* spermatocytes have a larger surface area and exhibit remarkable cell elongation. They also undergo morphological changes in intracellular membrane structures during late anaphase. The plasma membrane ingression occurs at the cleavage furrow sites in cytokinesis. Therefore, the vesicle transport system may play a critical role in male meiosis, and their impairment appears as a marked phenotype [23,24,57].

Cytokinesis was commonly perturbed in male meiotic cells harboring the depletion of either coat protein complex. The observation of the spermatid immediately after completion of meiosis II is a susceptible method that allows us to detect abnormalities in chromosome segregation and cytokinesis that occur even at low frequencies in two male meiotic divisions [24,57]. Nevertheless, only the cytokinesis phenotype was observed ([24], this study), with few cells derived from abnormal chromosome segregation or apoptotic cells. These observations indicate that COPI- and COPII-mediated vesicle transport is particularly important for cytokinesis in male meiosis. Mutants for the *bru* gene, encoding the membrane trafficking transport protein particle (TRAPP) II complex component, also exhibit abnormal cytokinesis in male meiosis but not neuroblast mitosis [29]. These phenotypes may be relevant because spermatocytes have a large surface area and are enriched in intracellular membrane structures [24,57]. 

COPII is required for vesicle transport from the ER to the Golgi and between the Golgi compartments. Although a few specific cargoes have been identified, many secreted proteins, or those localized in the plasma membrane and other organelles, are expected to be transported through the COPII-mediated pathways [58]. Sec24 is the primary COPII component responsible for cargo selection through either direct interaction with the ER exit signal on the cytoplasmic domain of the cargo proteins, which are transmembranes, or an indirect interaction through cargo receptors for soluble cargoes [34,59]. By contrast, COPI is responsible for intracellular transport from the Golgi back to the ER or between Golgi compartments [60]. Proteins in the Golgi lumen, which need to be transported to the ER, contain the signal peptide KDEL [61]. This sequence is recognized by a membrane-bound receptor, which binds to a guanine nucleotide exchange factor associated with ARF. It is possible to speculate that COPI and COPII convey different cargoes. This may explain differences in each depletion phenotype, i.e., whether they localize the CR components. A previous study reported that ER-based intracellular structures associated with astral and spindle microtubules are formed in the spermatocytes [23]. COPI depletion affects the central spindle microtubules, which are essential for cytokinesis. The authors propose that COPI plays a vital role in male meiosis through vesicle transport to the cleavage furrow region as well as the formation of ER-based structures. However, in COPII depletion, the CR components were appropriately localized at the onset of cytokinesis, and once the plasma membrane ingression initiated. Therefore, it is less likely that there were significant abnormalities in the ER-based structure and that they led to the inhibition of the central spindle microtubule formation.

## 4. Materials and Methods

### 4.1. Drosophila Stocks

For the silencing of mRNAs for COPII components and several other proteins, we used the following *UAS-RNAi* stocks: *P{TRiP.HMS00666}attP2* (#32878) from Bloomington Drosophila Stock Center (BDSC) (Indiana Univ., Bloomington, IN, USA) (hereafter, abbreviated as *UAS-Sec31RNAi*), *P{GD10593}v34191*(#10593) (*UAS-Sar1RNAi*), *P{KK102010}VIE-260B*(#102010)(*UAS-Sec16RNAi*), *P{GD4052}v37543*(#37543) (*UAS-Sec24CDRNAi*), *P{NIG.6773R}* (#6773R-1)(*UAS-Sec13RNAi*), *P{NIG.1250R}*(#1250R-1) (*UAS-Sec23RNAi*), and *P{NIG.1250R}*(#1472R-1) (*UAS-Sec24ABRNAi*) from the National Institute of Genetics (NIG) (Sizuoka, Japan). To generate germline cell clones homozygous for the lethal mutation for the *Sec24AB* (*Sec24AB^k06204^*) [62] by inducing somatic recombination using *hsp-FLP1* in the testes heterozygous for the mutation, we generated *p{ry[+t7.2]=neoFRT}42D Sec24AB^k06204^*/*p{ry[+t7.2]=neoFRT}42D p{w[+mc]=Ubi-mRFPnls}2R* male flies. To visualize COPII components, the related proteins, and several subcellular structures, the following stocks that induce fusion proteins with fluorescence tags were used: *P{UAS-EGFP.APEX2.Sec13}*, *P{UAS-Sec23.EGFP.APEX2}*, and *P{UAS-Sec31.EGFP.APEX2}* [39] (gifts from J. Parejya, USIH-UMH, Rome, Spain). *P{PLCγ-PH-eGFP}* (a gift from F. Pichaud (Univ. College London, London, UK)) was used to label the plasma membrane [63]; *P{Sqh-GFP*, *RLC}2* (a gift from R. Karess (CNRS, Paris, France)), to visualize Myosin Light Chain (MLC) [64]; *P{UASp-RFP.Golgi}5* (#30908, BDSC), to visualize the trans-Golgi apparatuses with RFP [65]; *P{Ubi-p63E-shg.GFP}* (#109007, Kyoto Drosophila Stock Center, Kyoto Institute of Technology, Kyoto, Japan), to recognize DE-cadherin; *P{βTub85D-mRFP-Anillin}* (a gift from J. Brill (The Hospital for Sick Children, Toronto, ON, Canada)), to visualize Anillin [42]; *P{Ubi-RFP-tub56D}* [66], to observe microtubules; and *P{Ubi-RFP-Sec13}* (in this study) was used to label the COPII vesicles containing Sec13 with RFP. To investigate the role of microtubules in the dynamics of the COPII vesicles, *asl*, *asl^2^* (gifts from S. Bonaccorsi (Univ. of Roma La Saplenza, Rome, Italy)), and *βTub85D^D^* (#107338, BDSC) were used. *P{bam-GAL4::VP16}* was used as a Gal4 driver for spermatocyte-specific gene expression [38]. We used *P{UAS-Dcr2}*; *P{bam-GAL4::VP16}* for spermatocyte-specific *RNAi* experiments to silence the relevant mRNAs [23]. All *Drosophila* stocks were maintained on standard cornmeal food at 25 °C as previously described [67]. For an efficient induction of GAL4-dependent gene expression, individuals carrying the GAL4 driver gene and UAS transgenes were raised at 28 °C. Fly food prepared according to a previous procedure was used to maintain the fly stocks [67].

### 4.2. Establishment of the Fly Stock to Express RFP-Tagged Sec13 

For the visualization of the COPII-coated vesicles during male meiosis, we established the *Drosophila* line expressing the RFP-tagged Sec13 protein. To construct the *p{UAS-RFP-Sec13}* plasmid using the GATEWAY Cloning System (Invitrogen, Tokyo, Japan), the full-length cDNA was amplified via PCR using the LD03471 plasmid (Drosophila Genomics Resource Center, Bloomington, IN, USA) as a template. The CACC sequence, which is a recognition sequence of topoisomerase, was added to the 5′ end of the forward primer (5′-CACCATGGTGAGCCTGCTGCAAGAG-3′) and reverse primer (5′-TCAGTTGGAGAGCTGCGAGTTGGAC-3′). The PCR product was cloned into the pENTRY/D-TOPO (Invitrogen, Tokyo, Japan) and then transferred into *Drosophila* Gateway vectors, pURW (Drosophila Genomics Resource Center, IN, USA), using the Gateway recombination protocol (Invitrogen, Waltham, MA, USA). After sequencing the resultant RFP-tagged protein expression plasmids, the plasmid DNA was microinjected (BestGene, Chino Hills, CA, USA) into the early embryos to establish the *UAS-RFP-Sec13* stock.

### 4.3. Quantitative Real-Time PCR (qRT-PCR) Analysis

Total RNA was extracted from 50 pairs of young adult testes using TRIzol reagent^®^ (Thermo Fisher Scientific, Waltham, MA, USA). cDNA synthesis using total RNA as a template was carried out using a Primer Script^®^ High Fidelity RT-PCR kit (TaKaRa, Shiga, Japan) with oligo dT primers. Real-time PCR was performed using FastStart Essential DNA Green Master (Roche, Mannheim, Germany) using the following: Rp49Fw: 5′-TTCCTGGTGCACAACGTG-3′, Rp49Rv: 5′-TCTCCTTGCGCTTCTTGG-3′, Sar1Fw: 5′-GGATACCTGGGTCTGTGGAA-3′, Sar1Rv:5′-GCCACCCAAGTCGAATGTAG-3′, Sec23Fw: 5′-GTGTCGGATGTGGAGATCG-3′, Sec23Rv: 5′-CATGCTGGTTGACCACCTC-3′, Sec13Fw: 5′-CGGTAAACTCGGTGGACTTC-3′, Sec13Rv: 5′-TACTCCGTGTTGCAGGTGAG-3′, Sec31Fw:5′-GTCATCCAGAATGTGCAACG-3′, Sec31Rv: 5′-ACCGTTCTGGTTCTGATTCG-3′, Sec24ABFw: 5′-CACGAAATGACGCTGCTG-3′, Sec24ABRv: 5′-TCACGCACCAGTTCCTTG-3′. Each sample was triplicated on the PCR plate, and the final results were the average of the three biological replicates. For the quantification, the ∆∆Ct method was used to determine the differences between target gene expression relative to the reference *Rp49* gene expression. These qRT-PCR experiments confirmed the efficient depletion (less than 30% of the controls) of the relevant mRNAs in testes expressing dsRNAs against the mRNAs for the coatomers and Sar1.

### 4.4. Preparation of Post-Meiotic Spermatid Cysts

To judge whether two consecutive meiotic divisions were executed correctly, we observed nuclei in post-meiotic spermatids at the onion stage just after the completion of meiosis II under phase-contrast microscopy, as previously described [3,68]. A pair of testes from pharate adults or newly eclosed adult flies (0–1 day old) were dissected to isolate spermatocyte cysts in Testis buffer (183 mM KCl, 47 mM NaCl, 10 mM EDTA, pH 6.8) and covered with a coverslip (Matsunami, Osaka, Japan) to flatten the cysts. For the observation of the spermatids under a phase-contrast microscope, the cysts collected from testes were mildly flattened in Testis buffer under a cover slip. After removing the coverslips, we transferred them into 100% methanol for 3 min at −30 °C to fix the samples. Subsequently, they were rehydrated in PBS (137.0 mM NaCl, 2.7 mM KCl, 10.1 mM Na_2_HPO_4_·12H_2_O, 1.8 mM KH_2_PO_4_), and then the DNA was stained with DAPI. Normal spermatocytes undergo two meiotic divisions to form 64 spermatocytes simultaneously. When these two meiotic divisions are carried out properly, a single spermatocyte possesses a one-to-one ratio of nucleus to mitochondrial aggregates (Nebenkerns) after the completion of meiosis II. The nucleus and the Nebenkern were seen to be white and black spheres under the phase-contrast microscope, respectively. However, if cytokinesis did not take place during meiosis, multiple nuclei (two or four) and Nebenkerns larger than the nucleus were generated in a single spermatid. Samples were observed using a phase-contrast microscope (Olympus, Tokyo, Japan, model: IX81).

### 4.5. Immunostaining

Testis cells were fixed in ethanol at −30 °C for 10 min, followed by 3.7% formaldehyde for 7 min. The slides were permeabilized in PBST (PBS containing 0.01% Triton-X) for 10 min and blocked with 10% normal goat serum in PBS. The following primary antibodies were used at the dilutions described: rabbit polyclonal anti-Sec23 antibody (1:400, PA1-069A, Thermo Fischer Scientific, Waltham, MA, USA), rabbit polyclonal anti-GM130 (cis-Golgi Marker) (ab30637), and anti-RFP antibody (1:400, M204-3, Medical and Biological Laboratories, Tokyo, Japan). After incubating overnight at 4 °C, the fixed samples were repeatedly washed in PBS and subsequently incubated with anti-Rabbit or anti-Guinea pig IgG (H + L) conjugated with Alexa Fluor 488 or 555 (Thermo Fisher Scientific, Waltham, MA, USA). After incubation for 2 h at room temperature, the samples were washed in PBS. They were mounted with VECTASHIELD Mounting Medium with DAPI (Vector Laboratories, Burlingame, CA, USA) and observed under a fluorescent microscope (Olympus, Tokyo, Japan, model: IX81). Image acquisition was controlled using Metamorph software version 7.6 (Molecular Devices, SAN Jose, CA, USA).

### 4.6. In Situ Proximity Ligation Assay (PLA)

We performed a PLA to detect close interactions between Sec13 and Sec13 using a Duolink kit (Sigma-Aldrich, St. Louis, MO, USA) as previously described [69]. We used the mouse anti-Sec23 and rabbit anti-RFP antibodies to detect the complex containing Sec13 and Sec23, and mouse anti-RFP and rabbit anti-Sec23 antibodies to detect the complex containing Sec13 and Sec23. Negative control experiments were also performed to confirm that only anti-RFP antibody produced a few PLA-positive foci. Image acquisition was performed as described above.

### 4.7. Wheat Germ Agglutinin (WGA) Staining 

To visualize the plasma membrane insertion into the cleavage sites of spermatocytes at late anaphase I to telophase I, the testis cells were stained with Alexa488-labeled WGA [70]. The cells were dehydrated and fixed in ethanol at −30 °C for 10 min, and subsequently fixed in 3.7% formaldehyde at room temperature for 15 min. For the simultaneous immunostaining with other antibodies, the WGA staining was performed after incubation with the secondary antibody in the immunostaining experiment.

### 4.8. Time-Lapse Observation of Male Meiotic Cells 

A time-lapse observation of spermatocytes undergoing meiosis I was performed according to a previous protocol [68,69]. Testes from adult males were collected in a drop of BRB Buffer (80 mM PIPES, 1 mM MgCl_2_, 1 mM EGTA, pH 6.8). The adjacent epididymis and fat bodies attached to the testis were completely removed. The testes were transferred to a pair in a drop of BRB Buffer placed on a cover glass. After a drop of Liquid Paraffin light (#26144-85, Nakala tesque, Kyoto, Japan) was overlayed on the drop, the remaining BRB below the testis sample was removed. Fluorescence, time-lapse observations were performed using an inverted phase contrast microscope to observe the first meiotic division. For each 60 sec interval, GFP and RFP fluorescence images were sequentially captured with a CCD camera (Hamamatsu Photonics, Shizuoka, Japan). Image acquisition was controlled using Metamorph software (Molecular Devices). The specimens were observed under a fluorescent microscope (Olympus, Tokyo, Japan, model: IX81). 

### 4.9. Statistical Analysis

More than 20 pairs of testes were used to quantitate the number and intensity of fluorescence foci in spermatocytes for each genotype. The results were presented as bar graphs using Prism (Version 9, GraphPad Software, San Diego, CA, USA) [71]. Each dataset was first assessed using the F-test to determine equal or unequal variance as described [71]. We used the Welch’s *t*-test test to compare the two groups. Statistical significance is described in each figure legend as follows: * *p* < 0.05, ** *p* < 0.01, *** *p* < 0.001, and **** *p* < 0.0001. A *p*-value of 0.05 or less was considered statistically significant. 

## Figures and Tables

**Figure 1 ijms-25-04526-f001:**
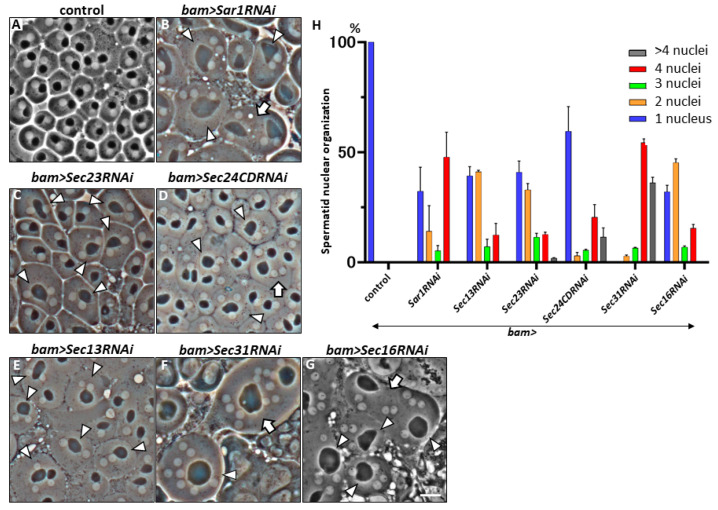
Phase-contrast images of spermatids derived from spermatocytes harboring the silencing of mRNAs for components consisting of COPII coat and regulators for its assembly. (**A**–**G**) Phase-contrast images of living spermatids at onion stage. (**A**) Control (*bam>+*) spermatids. Each cell contains a single nucleus (white) and a single Nebenkern, which is a mitochondrial aggregate (black) in a 1:1 ratio. The arrowheads indicate multinucleate cells harboring more than two nuclei and single Nebenkerns. The arrows indicate large cells in which two multinucleated cells have fused. (**B**) Spermatids that developed from *Sar1RNAi* spermatocytes (*bam>Sar1RNAi*). They possessed four or more than four nuclei with single Nebenkerns. (**C**) Spermatids derived from *Sec23RNAi* spermatocytes (*bam>Dcr-2*, *Sec23RNAi*). (**D**) Multinucleate spermatids derived from *Sec24CDRNAi* spermatocytes (*bam>Dcr-2*, *Sec24CDRNAi*). An arrow indicates a large cell in which two multinucleated cells have fused. (**E**) Spermatids derived from *Sec13RNAi* spermatocytes (*bam>Dcr-2*, *Sec13RNAi*). (**F**) Spermatids derived from *Sec31RNAi* spermatocytes (*bam>Sec31RNAi*). (**G**) Spermatids derived from *Sec16RNAi* spermatocytes (*bam>Sec16RNAi*). Bar, 10 µm. (**H**) Frequencies of control spermatids (single nuclei) and multinucleated spermatids (harboring two to more than four nuclei) at onion stage in the testes containing spermatocyte-specific silencing of COPII-related six genes. The bars represent the SEM, n > 107.

**Figure 2 ijms-25-04526-f002:**
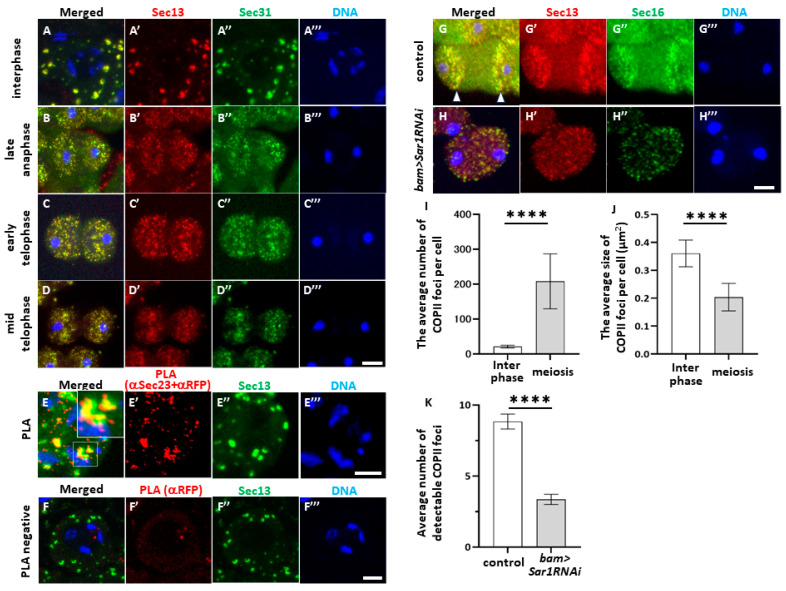
Co-localization and close association of two COPII components in pre-meiotic and meiotic spermatocytes. (**A**–**D**) Observation of spermatocytes expressing both RFP-Sec13 and Sec31-GFP before (**A**) and during meiosis I (**B**–**D**). The RFP fluorescence (red in (**A**–**D**), white in (**A**’–**D**’)). GFP fluorescence (green in (**A**–**D**) and (**A**’’–**D**’’)). DNA staining with DAPI (blue in (**A**–**D**) and (**A**’’’–**D**’’’). (**A**–**A**’’’) Spermatocyte at interphase. Yellow foci indicate co-localization of Sec13 and Sec31 (arrowheads). (**B**–**B**’’’) Spermatocyte at late anaphase. (**C**–**C**’’’) Spermatocyte at early telophase. (**D**–**D**’’’) Spermatocyte at mid-telophase. Scale bar: 10 µm. (**E**,**F**) In situ PLA signals (red in (**E**,**F**) and (**E**’,**F**’)) of normal spermatocytes expressing RFP-Sec13 (green in (**E**,**F**) and in (**E**”,**F**”)) to detect close association of Sec23 with Sec13. DNA staining with DAPI (blue in (**E**–**H**) and (**E**’’’–**H**’’’). (**E**) In situ PLA of the spermatocytes with both anti-Sec23 and anti-RFP antibodies, with magnified image of area shown in the inset. (**F**) Negative control for in situ PLA of cells with anti-RFP antibody. Few PLA signals appear in the spermatocytes. Scale bar: 10 μm. (**G**,**H**) Anti-Sec16 immunostaining (green in (**G**,**H**), white in (**G**’’,**H**’’)) of spermatocytes expressing RFP-Sec13 (red in (**G**,**H**) and (**G**’,**H**’)) in normal control (**G**) and *Sar1RNAi* spermatocyte (**H**) at anaphase I to telophase I. DNA staining with DAPI (blue in (**A**–**D**) and (**A**’’–**D**’’). Arrowheads indicate accumulation of COPII-containing foci in two cytoplasmic bands running from top to bottom of telophase cell. Bars: 10 μm. (**I**,**J**) Number (**I**) and size (**J**) of Sec13-positive foci in spermatocytes before meiosis (Interphase) and those undergoing meiosis I (meiosis). Control (*bam>RFP-Sec13*) (n = 29 cells) and *Sar1RNAi* (*bam>RFP-Sec13*, *Sar1RNAi*) (n = 40 cells). (**K**) Number of Sec13 foci in spermatocytes at telophase I. Control (*bam>RFP-Sec13*) (n = 44 cells) and *Sar1RNAi* cells (*bam>RFP-Sec13*, *Sar1RNAi*) (n = 51 cells). Bars represent the SEM. **** *p* < 0.0001 (Welch’s *t*-test).

**Figure 3 ijms-25-04526-f003:**
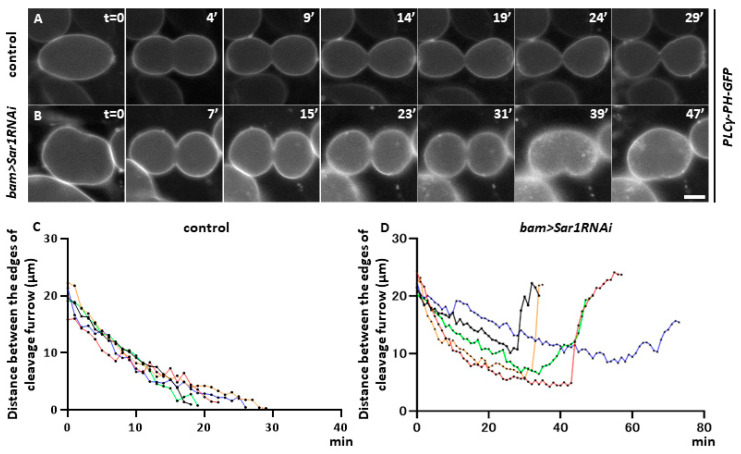
Time-lapse imaging of plasma membrane ingression during late anaphase I to the end of cytokinesis in male meiosis I. (**A**,**B**) Time-lapse observation of the plasma membrane labeled with GFP in meiosis I cells expressing *GFP-PLC*γ-*PH*. The observation started from the time when the ingression of the plasma membrane was initiated at the cell equator (t = 0′). (**A**) control cell (*bam>PLC*γ-*PH-GFP*). The membrane ingression terminates at 24 min after the start at CF sites. (**B**) The *Sar1*-silenced cells (*bam>PLC*γ-*PH-GFP*, *Sar1RNAi*). The ingression initiated at the cell equator of the silencing cell, terminates at the middle (t = 31′), and returns to its original state (t = 47′). Bar: 10 µm. (**C**,**D**) Time-lapse alteration in the distance between the edges of the CFs in control (*bam>+*) (**C**) and *Sar1*-silenced (*bam>Sar1RNAi*) (**D**) spermatocytes from late anaphase to the end of meiosis I. The recording was terminated at the point when the furrowing was completed or when no more changes in the cell diameter were observed in a control cell (*bam>PLC*γ-*PH-GFP*) or the *Sar1*-silenced cells (*bam>PLC*γ-*PH-GFP*, *Sar1RNAi*) (n = 5). The length of each cell is represented by a different color.

**Figure 4 ijms-25-04526-f004:**
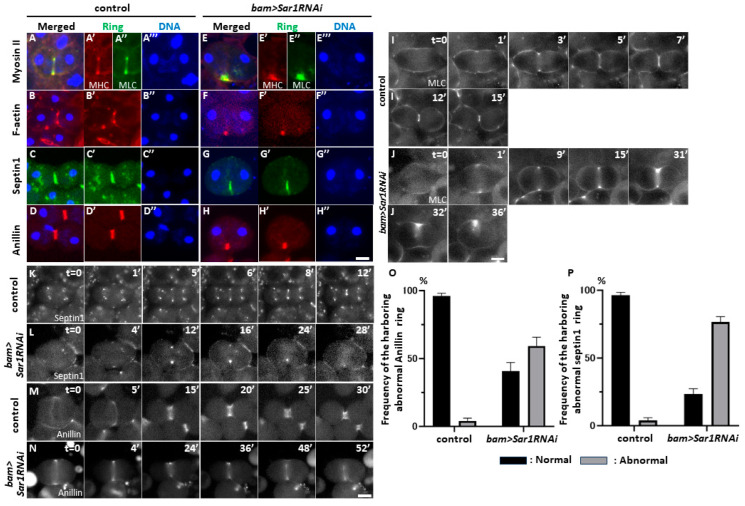
Abnormal localization of contractile ring components in *Sar1*-silenced spermatocytes at later stages of meiosis I. (**A**–**H**) Observation of the CRs using fluorescence-tagged CR proteins (Ring) in spermatocytes at late anaphase I to telophase I. (**A**,**E**) Immunostaining of the cells expressing GFP-MLC (green in (**A**,**A**’’,**E**,**E**’’)) with anti-Myosin Heavy Chain (MHC) antibody (red in (**A**,**A**’,**E**,**E**’)). (**B**,**F**) Fluorescence of the cells stained with Rhodamine-conjugated phalloidin to visualize F-actin (red in (**B**,**B**’,**F**,**F**’)). (**C**,**G**) Fluorescence of the cells expressing Septin 1-GFP (green in (**C**,**C**’,**G**,**G**’)). (**D**,**H**) Fluorescence of the cells expressing RFP-Anillin (red in (**D**,**D**’,**H**,**H**’)). (**A**–**D**) Normal control cells (*bam>+*). (**E**–**H**) *Sar1*-silenced cells (*bam>Sar1RNAi*). DNA staining with DAPI (blue in (**A**–**H**) and (**A**’’’, **B**”–**D**”, **E**’’’, **F**”–**H**”). (**I**,**J**) Time-lapse observation of contractile ring (CR) formation in the spermatocytes expressing MLC-GFP at anaphase I to cytokinesis in male meiosis. Control (**I**) and *Sar1*-silenced spermatocytes (**J**) were selected for visualization. In control cells, the MLC was accumulated at presumptive CF sites on the plasma membrane immediately before ingression (t = 0′), and the membrane ingression completed 15 min after the initiation of ingression. (**J**) In *Sar1*-silenced cells, the CR is constructed as in control cells (t = 36′). Thereafter, it was removed from the CF site on the lower plasma membrane, and the ingression returned to the initial state (t = 32′). The time-lapse observation was initiated at late anaphase I in both cells when the fluorescence of MLC-GFP became clearer (t = 0′) in both genotypes. (**K**–**N**) Time-lapse observation of the Septin1 CR (**K**,**L**) and the Anillin ring (**M**,**N**) from anaphase I just before the plasma membrane ingression at the cleavage furrow sites (white in Septin1-GFP) (t = 0 min) to the end of cytokinesis. Normal control spermatocyte (**K**,**M**) and *Sar1RNAi* spermatocyte (**L**,**N**). (**O**,**P**) Frequencies of the spermatocytes harboring an abnormal Anillin ring (**O**) or Septin1 ring (**P**) at late telophase I spermatocytes from control (*bam>+*) and *Sar1*-silenced (*bam>Sar1RNAi*) spermatocytes. The bars represent the SEM. Black bars: the telophase I cells harboring normal CRs. Gray bars: the telophase I cells harboring abnormally shaped CRs. Bars, 10 µm.

**Figure 5 ijms-25-04526-f005:**
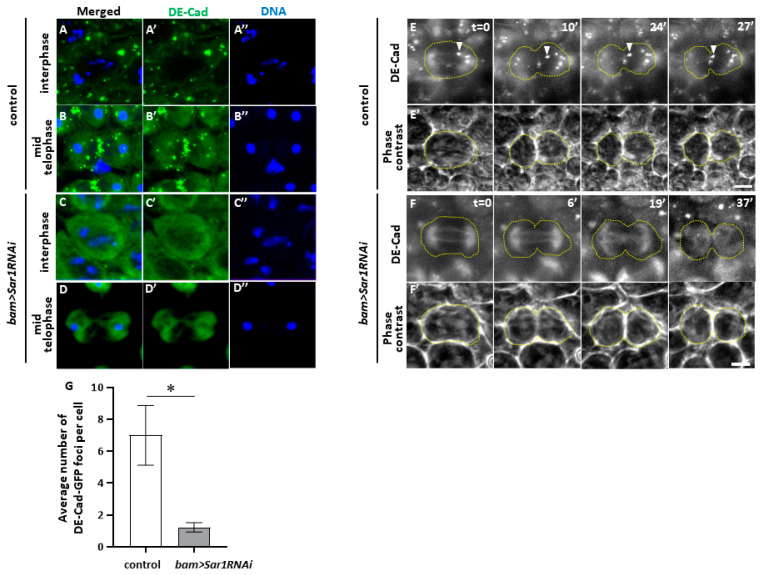
*Sar1*-dependent formation and distribution of DE-cad-GFP foci during the later stages of meiosis. (**A**–**D**) Observation of DE-cad-containing vesicles in the control (**A**,**B**) and *Sar1*-silenced (**C**,**D**) spermatocytes expressing DE-cad-GFP before meiosis (**A**,**C**) and at telophase I (**B**,**D**). DE-cad (green in (**A**–**D**) and (**A**’–**D**’)). DNA staining (blue in (**A**–**D**) and (**A**’’–**D**’’)). (**C**,**D**) Overexposed images. (**E**) Time-lapse observation of GFP fluorescence in living spermatocytes expressing GFP-DE-cad (white in (**E**)) during the later stages of meiosis I. As meiosis progresses, DE-cad-containing vesicles, indicated by arrowheads (t = 0′), migrated toward another vesicle at the cell equator (t = 24′–27′). (**F**) Time-lapse observation of DE-cad foci in living *Sar1*-silenced spermatocyte. According to the phase contrast micrographs of the cells (**E**’,**F**’), the cell margins are encircled by dotted lines. Arrows indicate a migration of the same vesicle in the periphery of a right spindle pole toward another vesicle at the cell equator as meiosis progresses. Note that few of the DE-cad vesicles were contained in the *Sar1*-silenced cell except for a weak and constant signal on the astral and spindle envelopes. (**G**) Average numbers of DE-cad-GFP foci in control (*bam>+*, *DE-cad-GFP*) (n = 58) and *Sar1*-silenced spermatocytes undergoing meiosis I (*bam>Sar1RNAi*, *DE-cad-GFP*) (n = 62). The bars represent the SEM. Significance was tested between control cells and the *Sar1*-silenced cells in meiosis I. * *p* < 0.05 (Welch’s *t*-test).

**Figure 6 ijms-25-04526-f006:**
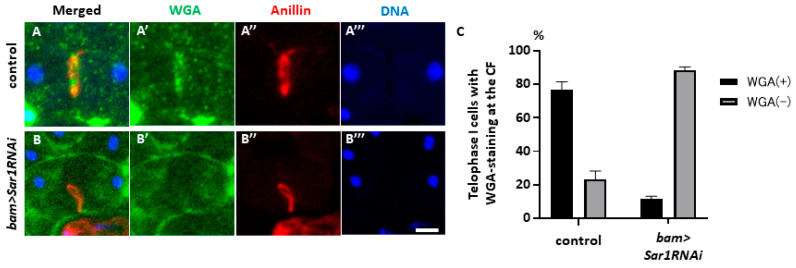
A loss of the plasma membrane newly added in the cleavage furrow by wheat germ agglutinin (WGA) in *Sar1*-silenced spermatocytes at telophase I. (**A**,**B**) A visualization of the plasma membrane stained with fluorescence-tagged WGA of the spermatocytes expressing RFP-tagged Anillin. The RFP-Anillin indicates the location of the CR in control (*bam>RFP-Anillin*) (**A**) and in *Sar1*-silenced cells (*bam>RFP-Sar1RNAi*, *RFP-Anillin*) (**B**) At telophase I. Note that the WGA signal is invisible in the cell midzone except for a weaker signal along the plasma membrane in the silenced cells, while the intense signal is mainly localized around the CR in control cells. WGA staining (green in (**A**,**B**), white in (**A**’,**B**’)), RFP-Anillin (red in (**A**,**A**’’,**B**,**B**’’)), and DNA (blue in (**A**,**A**’’’,**B**,**B**’’’)). (**C**) Frequencies of cells exhibiting no or reduced WGA signals (gray bars) and distinctive signals (black bars) in the midzone among the telophase I cells in control and the *Sar1*-silenced cells (n = 36 telophase I cells in total in both control and the silenced cells). The bars represent the SEM.

## Data Availability

The datasets generated and/or analyzed in the current study are available from the corresponding author upon reasonable request.

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
