# Peer review of "Essential Role of COPII Proteins in Maintaining the Contractile Ring Anchoring to the Plasma Membrane during Cytokinesis in Drosophila Male Meiosis"

_ijms, 2024, doi:10.3390/ijms25084526_

Round 1

Reviewer 1 Report

Comments and Suggestions for Authors

This manuscript by Matsuura et al. describes a study of the role of COPII proteins in cytokinesis during Drosophila spermatogenesis. The authors found that knock down of COPII components causes cytokinesis failure during meiotic cell divisions, resulting in multinucleated spermatids. They document that the cleavage furrow is initiated, but fails to progress. The authors further explore possible subcellular transport defects that lead to this phenotype (E-Cadherin, Septin1, lipid droplets). However, no convincing case is made that COPII-mediated transport of any of these factors is a direct explanation for the phenotype (see my comments below). For this, I found the study too preliminary for publication, and its main conclusions and title unsupported by the evidence. If the authors insist they wish to publish this, they should discard Figs 5, 6 and 7 (inconclusive/preliminary/unsupported), leave the description of the phenotype and COPII localizations (Figs 1-4) and finish the paper with Fig 8 (involvement of the secretory pathway in the arrival of new membrane to the cleavage furrow), which seems convincing enough and a starting point for future studies.

Main comments:

1. The authors show that overexpressed DE-cadherin-GFP arrives to the cleavage furrow in a sar1-dependent manner (Fig 5). This is convincing. However, DE-cadherin has no role in cytokinesis and the significance of the reported rescue is difficult to interpret. The only way I see these data are useful is by saying DE-cadherin-GFP, as a transmembrane protein, is a marker of secretory membrane arrival to the cleavage furrow and include panels 5A-G in Fig 8.

2. The authors show that overexpressed Septin 1-GFP localizes in ER-Golgi (Fig 6). This is surprising and very suggestive. The authors should confirm this with an antibody for endogenous septin or Pnut. But even if confirmed, septin is a cytoplasmic protein and COPII vesicles can’t have a role in their transport to the cleavage furrow. In fact, the authors report that Septin1 arrives at the cleavage furrow in a sar1-independent way (not affected by sar1 knock down). Therefore, I believe this is an interesting observation that doesn’t belong in this study. I recommend the authors save it and explore it further. If they insist in including it here, reporting localization of endogenous septin is essential.

3. The authors show differences in nile red staining with the wild type in sar1 knock down (Fig 7) and conclude and involvement of COPII in the transport of lipid droplets to the cleavage furrow. First, not all this Nile red signal can be neutral lipids or lipid droplets; there is plenty of it in the plasma membrane, including the cleavage furrow. Second, the difference with the wild type is lack of ER-associated nile red signal (membrane or lipid droplet), rather than failure of transport to the cleavage furrow. Third, the authors seem to think the notion that lipid droplets can contribute directly to cleavage furrow membrane incorporation is obvious (perhaps by fusing with the plasma membrane), but this is extremely unlikely for all we know about lipid droplets. 

4. This same laboratory previously reported that COPI knock down resulted in multinucleate spermatids (Kitazawa et al., 2012). On that occasion, they attributed the defect to defects in ER-spindle relations. Given that COPI and COPII loss usually produce similar effects, the authors should discuss that previous work in light of this new study.

Other comments:

Line 14: Four subunits

Line 68: COPI = Golgi-ER retrograde

Line 70: COPII = ER-Golgi anterograde

Line 123: revise sentence

Comments on the Quality of English Language

Minor editing

Author Response

Reviewer 1

This manuscript by Matsuura et al. describes a study of the role of COPII proteins in cytokinesis during Drosophila spermatogenesis. The authors found that knock down of COPII components causes cytokinesis failure during meiotic cell divisions, resulting in multinucleated spermatids. They document that the cleavage furrow is initiated, but fails to progress. The authors further explore possible subcellular transport defects that lead to this phenotype (E-Cadherin, Septin1, lipid droplets). However, no convincing case is made that COPII-mediated transport of any of these factors is a direct explanation for the phenotype (see my comments below). For this, I found the study too preliminary for publication, and its main conclusions and title unsupported by the evidence. If the authors insist they wish to publish this, they should discard Figs 5, 6 and 7 (inconclusive/preliminary/unsupported), leave the description of the phenotype and COPII localizations (Figs 1-4) and finish the paper with Fig 8 (involvement of the secretory pathway in the arrival of new membrane to the cleavage furrow), which seems convincing enough and a starting point for future studies.

Main comments:

  1. The authors show that overexpressed DE-cadherin-GFP arrives to the cleavage furrow in a sar1-dependent manner (Fig 5). This is convincing. However, DE-cadherin has no role in cytokinesis and the significance of the reported rescue is difficult to interpret. The only way I see these data are useful is by saying DE-cadherin-GFP, as a transmembrane protein, is a marker of secretory membrane arrival to the cleavage furrow and include panels 5A-G in Fig 8.

We understand the reviewer’s concern that the downregulation of DE-cadherin showed no phenotype in cytokinesis, and that the rescue of the Sar1 depletion phenotype by overexpression of DE-Cadherin is difficult to interpret. According to the review’s request, we removed 5H from the manuscript but left panels 5A-G. We agreed with the reviewer’s suggestion and described only the dynamics that the vesicle containing DE-Cadherin, which is a marker of transmembrane proteins, was constructed dependent on Sar1 and transported to the cleavage sites. We also presented panels A-G in the previous Figure 8 as the new Figure 6.

In addition, in response to the reviewer's criticism that “title is unsupported by the evidence”, we also revised the title as follows: “Essential Role of COPII Proteins in Maintaining the Contractile Ring anchoring to the Plasma Membrane during Cytokinesis in Drosophila Male Meiosis”.

  1. The authors show that overexpressed Septin 1-GFP localizes in ER-Golgi (Fig 6). This is surprising and very suggestive. The authors should confirm this with an antibody for endogenous septin or Pnut. But even if confirmed, septin is a cytoplasmic protein and COPII vesicles can’t have a role in their transport to the cleavage furrow. In fact, the authors report that Septin1 arrives at the cleavage furrow in a sar1-independent way (not affected by sar1 knock down). Therefore, I believe this is an interesting observation that doesn’t belong in this study. I recommend the authors save it and explore it further. If they insist in including it here, reporting localization of endogenous septin is essential.

We have performed the immunostaining of the male meiotic cells at telophase I with the anti-Septin 1 antibody (Mol. Biol. Cell 26:15-28, 2015) . However, we have not succeeded to visualize the Septin 1-containing vesicles in the cytoplasm, although it recognized the contractile ring containing the Septin 1 at the cleavage sites. We suspected that the antibody could not recognize the Septin 1 stored in the vesicles, even if it could recognize the protein in the contractile ring. As we agree the reviewer’s request that these data concerning the Septin 1 are  removed from the current manuscript to prepare another paper together with some additional data.

  1. The authors show differences in nile red staining with the wild type in sar1 knock down (Fig 7) and conclude and involvement of COPII in the transport of lipid droplets to the cleavage furrow. First, not all this Nile red signal can be neutral lipids or lipid droplets; there is plenty of it in the plasma membrane, including the cleavage furrow. Second, the difference with the wild type is lack of ER-associated nile red signal (membrane or lipid droplet), rather than failure of transport to the cleavage furrow. Third, the authors seem to think the notion that lipid droplets can contribute directly to cleavage furrow membrane incorporation is obvious (perhaps by fusing with the plasma membrane), but this is extremely unlikely for all we know about lipid droplets.

A previous paper from our laboratory reported that the COPI depletion resulted in the failure of cytokinesis through disrupted accumulation of lipid droplets labelled by Bodipy 493/503 (Molecular Probe) as well as that of essential proteins for cytokinesis. As the same fluorescence dye was not available in this time, we used the Nile Red instead. Taking carefully into account the three criticisms by the reviewer, particularly the third point, we removed section 2.9 in the Results, Figure 7, and the relevant sentences that were present in the previous 3.5 in Discussion (lines 662-665, 668-671).

  1. This same laboratory previously reported that COPI knock down resulted in multinucleate spermatids (Kitazawa et al., 2012). On that occasion, they attributed the defect to defects in ER-spindle relations. Given that COPI and COPII loss usually produce similar effects, the authors should discuss that previous work in light of this new study.

At the end of the Discussion, we added the section 3.5 to discuss the shared roles of these vesicle transport systems mediated by these two coatomer protein complexes in Drosophila male meiosis, and the differences between them, in addition to the differences in their function in mammalian mitosis. We also argued as follows (line 577-585) “A previous study reported that ER-based intracellular structures associated with astral and spindle microtubules are formed in the spermatocytes [23]. It was argued that the COPI depletion resulted in a reduction in the number of overlapping central spindle microtubules, which are essential for cytokinesis. The authors proposed that COPI plays a vital role in Drosophila male meiosis, not only through vesicle transport to the cleavage furrow region, but also through the formation of ER-based structures. However, in the COPII depletion, the CR components were appropriately localized at the onset of cytokinesis, and the plasma membrane ingression once initiated. Therefore, it is less likely that there were significant abnormalities in the ER-based structure and that they led tothe inhibition of the central spindle microtubule formation.”

Other comments:

Line 14: Four subunits

We corrected the mistake accordingly (line 17).

Line 68: COPI = Golgi-ER retrograde

We revised the phrase as requested (lines 66-67).

Line 70: COPII = ER-Golgi anterograde

We revised the phrase as requested (line 69).

Line 123: revise sentence

We revised the first subtitle of the Results as follows: “Multi-Nuclear Cells in the Spermatid Cysts derived from Spermatocyte That Harbored Silencing of mRNAs Encoding COPII Coatomer Proteins” (lines  121-122)

Reviewer 2 Report

Comments and Suggestions for Authors

Upon nucleus division -karyokinesis- a cell undergoes cytoplasm division -cytokinesis-. The process is tightly dependent on the actin-myosin component, which forms a contractile ring localized underneath the plasma membrane at the equatorial region of the cell. The latter creates a cleavage furrow responsible for the ingression of the plasma membrane. The cytokinesis process is evolutionary conserved and myosin has been shown to play a crucial role. Afterward, the two daughter cells pull apart forming a thin midbody. In conjunction with the actomyosin cytoskeletal components, at the cleavage furrow level, the requirement of new membrane components is crucial. This is important for the plasma membrane ingression at the cleavage furrow site, and also in this case the process appears to be evolutionary conserved in animals, including the toad Xenopus and the worm C. elegans. The COP complexes, COPI and COPII, are crucial players in intracellular vesicular transport. Whereas COPI regulates the so-called retrograde transport, COPII is involved in controlling the anterograde transport. While the role of COPI in controlling the cytokinesis of male meiosis has been explored in Drosophila, that of COPII is so far unexplored.

The authors of the study titled "Essential Role of COPII Proteins in Transporting Factors for Anchoring the Contractile Ring in the Plasma Membrane During Cytokinesis in Drosophila Male Meiosis" by employing the genetically amenable Drosophila model explore the role of the COPII  in male meiosis.

Overall the inquiry is well articulated and the findings support the conclusion, though DE-Cad point presents some flaws recognized by the authors themselves (page 17 lines 627-633).

Before publication, authors are asked to provide some clarification on the small issues below shortly discussed.

Minor issues

  • Concerning the figures (immunofluorescence, time-lapse) it would be warmly appreciated if the authors might provide the single channels images (i.e., red and green) instead of the black and white images. Currently, just the merge panel is colored.

  • What does encode for the asterless gene?

  • Page 10, lines 346-347, and the following issue throughout the manuscript: I am not fully sure that "detached" describes properly what occurs. Perhaps, more suitable could be mis-localized or re-localized. My concern comes because when such a complex detaches from the plasma membrane becomes soluble and to robustly prove that, one would need some biochemical experiments separating the plasma membrane from the cytoplasm and then assessing the localization through Western blot.

  • When it comes to the overexpression of De-Cad and arm it seems that the phenotype is rescued, but only partly. Why not assess whether overexpressing both together, the outcome would be a full rescue?

  • I might have missed something, and if so I apologize for that, but to me, it is not clear why the authors describe qRT-PCR in the section "Materials and Methods" when I did not find any mRNA quantification. Could you please clarify the issue?

  • Quite a few typos (e.g., line 54: Xenopus italics as well in the other part of the manuscript; line 84: why is Arf6 in brackets?; line 220: I guess that "sciented" should be silenced, is it not?; lines 322-323: if the MHC acronym is detailed in extenso, for consistency I would suggest to the authors to do the same for MLC, which I guess to be Myosin Light Chain, is it not?; line 819: there seem to be an extra bracket, and few others typos) are scattered throughout the main text.
Comments on the Quality of English Language

The language requires minor editing and a few typos are scattered throughout the manuscript, as detailed in the comments and suggestions for authors.

Author Response

Reviewer2

Upon nucleus division -karyokinesis- a cell undergoes cytoplasm division -cytokinesis-. The process is tightly dependent on the actin-myosin component, which forms a contractile ring localized underneath the plasma membrane at the equatorial region of the cell. The latter creates a cleavage furrow responsible for the ingression of the plasma membrane. The cytokinesis process is evolutionary conserved and myosin has been shown to play a crucial role. Afterward, the two daughter cells pull apart forming a thin midbody. In conjunction with the actomyosin cytoskeletal components, at the cleavage furrow level, the requirement of new membrane components is crucial. This is important for the plasma membrane ingression at the cleavage furrow site, and also in this case the process appears to be evolutionary conserved in animals, including the toad Xenopus and the worm C. elegans. The COP complexes, COPI and COPII, are crucial players in intracellular vesicular transport. Whereas COPI regulates the so-called retrograde transport, COPII is involved in controlling the anterograde transport. While the role of COPI in controlling the cytokinesis of male meiosis has been explored in Drosophila, that of COPII is so far unexplored.

The authors of the study titled "Essential Role of COPII Proteins in Transporting Factors for Anchoring the Contractile Ring in the Plasma Membrane During Cytokinesis in Drosophila Male Meiosis" by employing the genetically amenable Drosophila model explore the role of the COPII  in male meiosis.

Overall the inquiry is well articulated and the findings support the conclusion, though DE-Cad point presents some flaws recognized by the authors themselves (page 17 lines 627-633).

Before publication, authors are asked to provide some clarification on the small issues below shortly discussed.

Minor issues

  1. Concerning the figures (immunofluorescence, time-lapse) it would be warmly appreciated if the authors might provide the single channels images (i.e., red and green) instead of the black and white images. Currently, just the merge panel is colored.

According to the reviewer's request, we replaced black and white images of the immunofluorescence images (Figs. 2A'-A"', 2B'-B"', 2C'-C"', 2D'-D"', 2E'-E"'E, 2F'-F"'F, 2G'-G"', 2H'-H"', 4A'-A"', 4B'-B"', 4C'-C"', 4D'-D"', 5A'-A", 5B'-B", 5C'-C", 5D'-D", 6A'-A"', and 6B'-B"').

  1. What does encode for the asterless gene?

The asterless gene product is essential in astral microtubule formation during Drosophila male meiosis (Bonaccorsi et al., 1998). Therefore, we used this asterless mutant male to investigate whether the distribution of the COPII-containing foci was affected without proper astral microtubules.

  1. Page 10, lines 346-347, and the following issue throughout the manuscript: I am not fully sure that "detached" describes properly what occurs. Perhaps, more suitable could be mis-localized or re-localized. My concern comes because when such a complex detaches from the plasma membrane becomes soluble and to robustly prove that, one would need some biochemical experiments separating the plasma membrane from the cytoplasm and then assessing the localization through Western blot.

As the reviewer pointed out, we initially observed the contractile rings in the fixed cells at telophase I and found that these rings were abnormally localized on one side of the cleavage furrow sites in the Sar1-depleted cells. The only conclusion that can be drawn from this observation is that the ring components were mis-localized by inhibition of the COPII vesicle formation, as the reviewer mentioned. However, we further performed a time-lapse observation to follow up the dynamics of the ring from late anaphase to the end of cytokinesis. As shown in Fig. 4I, J, the MLC ring was once formed on the CF sites at both cell sides. Subsequently, it seemed to “be removed” from the lower side of the membrane in the Sar1-depleted cells at or before t=32’ after the onset of CF ingression. Consistently, we found a similar detachment of the Septin 1 and Anillin rings that were once anchored to the membrane upon the ring constriction. Other papers used the word “detached” or “slipped” along the cell membrane when they described the same phenotype that the contractile rings come off from the plasma membrane upon myosin ring contraction  (Baldauf, et al. 2022, ACS Synth Biol. 11:3120-3133). However, following the reviewer’s comment, we replaced the word “detached” with “removed”.  As the reviewer expected, the MLC-GFP signal dispersed around the CF site and disappeared shortly after the ring detachment (t=32’~36’). We agree that a biochemical fractionation analysis is more suitable to show the shift of MLC from the insoluble membrane fraction to the soluble cytoplasmic fraction. We should address this issue in our subsequent study. We appreciate the reviewer’s valuable suggestion.

  1. When it comes to the overexpression of De-Cad and arm it seems that the phenotype is rescued, but only partly. Why not assess whether overexpressing both together, the outcome would be a full rescue?

As instructed by reviewer 1, we removed the data concerning the genetic interaction between the Sar1 and the DE-Cadherin gene or arm to prepare a different paper. However, this is an exciting idea. As it will take at least a few months (several generations) to establish the fly stocks that allow us to examine the issue, we will continue the experiment and report the results in our subsequent paper.

  1. I might have missed something, and if so I apologize for that, but to me, it is not clear why the authors describe qRT-PCR in the section "Materials and Methods" when I did not find any mRNA quantification. Could you please clarify the issue?

We performed the qRT-PCR to confirm the efficient depletion of each mRNA encoding the COPII component. We added the following sentence at line 699 : “These qRT-PCR experiments confirmed the efficient depletion (less than 30% of the controls) of the relevant mRNAs in testes expressing dsRNAs against the mRNAs for those coatomers and Sar1.”

  1. Quite a few typos (e.g., line 54: Xenopus italics as well in the other part of the manuscript; line 84: why is Arf6 in brackets?; line 220: I guess that "sciented" should be silenced, is it not?; lines 322-323: if the MHC acronym is detailed in extenso, for consistency I would suggest to the authors to do the same for MLC, which I guess to be Myosin Light Chain, is it not?; line 819: there seem to be an extra bracket, and few others typos) are scattered throughout the main text.

We appreciate the reviewer’s careful reading of our manuscript. According to the reviewer’s request, we revised the word of Xenopus to italic (line 49, 52). We also corrected the typo concerning brackets containing Arf6 (line 82). Regarding the immunofluorescence images in Fig. 4A, we performed anti-MHC (Myosis Heavy Chain) immunostaining of spermatocytes expressing the GFP-tagged MLC (Myosis Light Chain). We believed that the word of MHC was correct (line 318). As requested, we removed the extra brackets (line 716).

Round 2

Reviewer 1 Report

Comments and Suggestions for Authors

My concerns have been addressed. 

Comments on the Quality of English Language

Minor editing of English language required